# A Retrospective Comparison of Three Patient-Controlled Analgesic Strategies: Intravenous Opioid Analgesia Plus Abdominal Wall Nerve Blocks versus Epidural Analgesia versus Intravenous Opioid Analgesia Alone in Open Liver Surgery

**DOI:** 10.3390/biomedicines10102411

**Published:** 2022-09-27

**Authors:** Hsin-I Tsai, Yu-Chieh Lu, Chih-Wen Zheng, Ming-Chin Yu, An-Hsun Chou, Cheng-Han Lee, Hao-Wei Kou, Jr-Rung Lin, Yu-Hua Lai, Li-Ling Chang, Chao-Wei Lee

**Affiliations:** 1Department of Anesthesiology, Chang Gung Memorial Hospital, Linkou Medical Center, Taoyuan 333, Taiwan; 2College of Medicine, Chang Gung University, Taoyuan 333, Taiwan; 3Graduate Institute of Clinical Medical Sciences, Chang Gung University, Taoyuan 333, Taiwan; 4Department of Surgery, Chang Gung Memorial Hospital, Linkou Medical Center, Taoyuan 333, Taiwan; 5Department of Surgery, New Taipei Municipal Tu-Cheng Hospital (Built and Operated by Chang Gung Medical Foundation), Tu-Cheng, New Taipei City 236, Taiwan; 6Department of Gastroenterology and Hepatology, Chang Gung Memorial Hospital, Linkou Medical Center, Taoyuan 333, Taiwan; 7Division of General Surgery, Department of Surgery, Chang Gung Memorial Hospital, Linkou Medical Center, Taoyuan 333, Taiwan; 8Clinical Informatics and Medical Statistics Research Center and Graduate Institute of Clinical Medicine, Chang Gung University, Taoyuan 333, Taiwan; 9Department of Nursing, Chang Gung Memorial Hospital, Linkou Medical Center, Taoyuan 333, Taiwan

**Keywords:** patient-controlled epidural analgesia, intravenous analgesia, nerve block, liver resection, hepatectomy

## Abstract

**Background**: Adequate pain control is of crucial importance to patient recovery and satisfaction following abdominal surgeries. The optimal analgesia regimen remains controversial in liver resections. **Methods**: Three groups of patients undergoing open hepatectomies were retrospectively analyzed, reviewing intravenous patient-controlled analgesia (IV-PCA) versus IV-PCA in addition to bilateral rectus sheath and subcostal transversus abdominis plane nerve blocks (IV-PCA + NBs) versus patient-controlled thoracic epidural analgesia (TEA). Patient-reported pain scores and clinical data were extracted and correlated with the method of analgesia. Outcomes included total morphine consumption and numerical rating scale (NRS) at rest and on movement over the first three postoperative days, time to remove the nasogastric tube and urinary catheter, time to commence on fluid and soft diet, and length of hospital stay. **Results**: The TEA group required less morphine over the first three postoperative days than IV-PCA and IV-PCA + NBs groups (9.21 ± 4.91 mg, 83.53 ± 49.51 mg, and 64.17 ± 31.96 mg, respectively, *p* < 0.001). Even though no statistical difference was demonstrated in NRS scores on the first three postoperative days at rest and on movement, the IV-PCA group showed delayed removal of urinary catheter (removal on postoperative day 4.93 ± 5.08, 3.87 ± 1.31, and 3.70 ± 1.30, respectively) and prolonged length of hospital stay (discharged on postoperative day 12.71 ± 7.26, 11.79 ± 5.71, and 10.02 ± 4.52, respectively) as compared to IV-PCA + NBs and TEA groups. **Conclusions**: For postoperative pain management, it is expected that the TEA group required the least amount of opioid; however, IV-PCA + NBs and TEA demonstrated comparable postoperative outcomes, namely, the time to remove nasogastric tube/urinary catheter, to start the diet, and the length of hospital stay. IV-PCA with NBs could thus be a reliable analgesic modality for patients undergoing open liver resections.

## 1. Introduction

Enhanced recovery after surgery (ERAS) protocol has been well developed and practiced since the 1990s in various surgical subspecialties [1,2,3,4,5,6,7]. Among the ERAS protocol, postoperative analgesia is an integral part that can reduce wound pain, facilitate earlier mobilization, and enhance the satisfaction of the patients effectively. Patients receiving conventional open hepato-biliary surgeries, among others, were particularly vulnerable to wound pain since they usually present with multiple comorbidities and are left with a large incision [8]. The most frequently used surgical incision for open liver resection is modified Makuuchi incision, which commences from xiphoid to about one to two centimeters above the umbilicus, and then extends laterally to the right [9,10,11,12]. According to the ERAS society, there is clinical equipoise in postoperative pain control for open liver surgery [13]. Adequate postoperative pain control is imperative for early mobilization and enhanced respiratory function. Intravenous patient-controlled analgesia (IV-PCA), anterior abdominal wall nerve blocks (NBs), and thoracic epidural analgesia (TEA) [14] have been commonly utilized. Among them, TEA provides opioid-free analgesia with fewer respiratory complications. Although serious complications are rare, the routine use of epidural analgesia may be limited by its significant failure rate (20–30%) and its potential to lead to hypotension requiring vasopressors. Other severe complications include epidural hematoma (0.02%) and abscess (0.1%), postoperative kidney failure, and neurologic deficits (1.12%) [15,16,17,18,19,20,21,22,23,24]. The application of TEA was not routinely recommended in open liver surgery as a result [13]. When TEA is contraindicated, such as patient refusal, coagulation abnormalities, elevated intracranial pressure, and infection over the Tuohy needle insertion site, IV-PCA appears to be a reasonable alternative for pain control [25]. Nonetheless, opioid is often associated with adverse drug events (ADEs), including nausea, vomiting, delayed return of urinary and bowel function, over-sedation, respiratory depression, and exacerbation of hepatic encephalopathy [26,27], which becomes significant postoperatively when opioid metabolism may be impaired in correlation with the size of liver resection [28]. Optimal pain control without severe ADEs, therefore, is of utmost importance. Recently, regional NBs of the anterior abdominal wall, for example, transversus abdominis plane (TAP) and rectus sheath (RS) blocks, have played a role in the multimodal analgesia regimens. These blocks appear promising as analgesia is achieved while reducing opioid requirements. Patients may be able to have improved respiratory function leading to a shorter length of hospital stay and better satisfaction of the patients [26,27,29,30,31,32,33]. Generally, an open hepatectomy incision requires dermatome coverage from approximately T6 to T10, which can be covered by bilateral TAP blocks or left-sided rectus sheath block in addition to a right-sided TAP block [28]. Although the TAP block has been reported with complications such as systemic toxicity associated with local anesthetics [34], liver injury [35,36], and transient nerve palsy under a blind approach [37], the advent of modern technology has reduced these unfavorable events to a minimum. When performed under ultrasound guidance, anterior abdominal wall NBs can be achieved with direct observation of needle placement and the spread of local anesthetics into the correct plane. As controversies in choosing the optimal pain management modality in patient’s receiving liver resection still exist, we aimed to compare three analgesia methods, IV-PCA, IV-PCA in addition to NBs, and patient-controlled TEA (PCEA) for postoperative pain control following open liver resections.

## 2. Material and Methods

### 2.1. Data Source and Patient Population

Under the approval of the Institutional Review Boards of Chang Gung Memorial Hospital (CGMH) (CGMH IRB No: 202000571B0), we retrospectively acquired data from the CGMH Pain Service database that included patient demographics, diagnosis of disease, surgical procedures, medication, and postoperative-adverse effects. From January 2015 to December 2019, 243 patients undergoing elective hepatectomy were retrospectively selected for the study. All data were fully anonymized and the requirement for informed consent was waived by the CGMH IRB committee. To avoid bias in the practice of surgery and postoperative care, only patients from one surgical team in the Division of General Surgery, Department of Surgery, Linkou CGMH were included. All patients received the modified Makuuchi incision, and the operative techniques as well as the postoperative management were described in our previous publications [38,39]. The demographics of the patients included age, gender, weight, preoperative comorbidities, alcohol or illicit drug abuse, and cigarette smoking. Other relevant surgical history included, in addition to common intraoperative parameters, the indication and extent of liver resection, the presence of intraabdominal adhesion, and the presence of distant metastasis. The exclusion criteria included patients of other surgical teams and those who received laparoscopic hepatectomy. Patients were allocated into one of the three groups according to the type of analgesic modalities chosen at the time of surgery: IV-PCA *(n* = 85), IV-PCA + NBs (*n* = 114), and TEA (*n* = 44), as demonstrated in Figure 1.

### 2.2. Procedures of Anesthesia and Postoperative Analgesia

All patients received a standardized general anesthetic technique for induction with intravenous (IV) 50–100 mcg fentanyl, 2–3 mg/kg propofol, and 0.2 mg/kg cisatracurium and 20–40 mg lidocaine hydrochloride. Maintenance was achieved with oxygen/air sevoflurane and IV fentanyl. Upon arriving post-anesthetic care unit (PACU), one of three postoperative analgesic strategies was commenced. (1) IV-PCA: patients received 1 mcg/kg IV fentanyl bolus dose before IV-PCA was started at a basal rate of 1–2 mL/h, demand bolus 2–4 mL, and lockout interval time of 5 min. (2) IV-PCA + NBs: patient received IV-PCA and bilateral RS [40] and right-sided subcostal TAP nerve blocks under ultrasound (US) guidance [41]. Nerve blocks were performed under an aseptic technique using a US transducer (linear 6–13 MHz, SonoSite M-Turbo, Brothell, WA, USA). A 23-gauge 70 mm needle (NIPRO Co., Shanghai, China) was inserted in-plane to the transducer medial to lateral direction with the endpoint in the fascial plane between the rectus muscle and posterior RS or between the rectus abdominis and the transversus abdominis muscle. For NBs, 20 mL of 0.33% ropivacaine with dexamethasone [42] were used for each injection. Following negative aspiration, 20 mL of 0.33% ropivacaine was injected with intermittent aspiration every 5 mL while observing the expansion of the intermuscular plane. In this group, IV-PCA was started at a basal rate 0.5–1 mL/h, demand bolus 1–3 mL and lockout interval time of 5 min. IV-PCA solution was constituted of fentanyl 500 μg and morphine 40 mg in 236 mL of normal saline to a total of 340 mL, giving a concentration of 1.47 μg/mL of fentanyl and 0.12 mg/mL of morphine. (3) TEA: patients received 1 mcg/kg IV fentanyl and 2 mg midazolam during epidural catheter insertion for patient-controlled epidural analgesia (PCEA). PCEA solution was constituted of 300 μg fentanyl and 600 mg bupivacaine in 474 mL normal saline to a total of 600 mL. PCEA was set at a basal rate of 4–6 mL/h, demand bolus 4–6 mL, and lockout interval of 20 min. All patients were given instructions on how to use PCA and followed up for the next three postoperative days. For the next three days, the acute pain service personnel would record the cumulative consumption of PCA, pain intensity based on the numeric rating scale (NRS) at rest and upon movement and the presence of opioid-related side effects [43,44,45]. The rescue analgesia was given as per hospital policy in all three groups if the pain score (NRS ≥ 4). For the convenience of comparison of opioid requirements, the total amount of opioids consumed in the post-anesthesia care unit (PACU) and on the ward over a 3-day period was converted to their equivalent morphine dose in milligrams using the equianalgesic conversion ratios of fentanyl: morphine = 0.1:10 [46,47].

Five major outcomes were assessed following hepatectomy, namely total consumption of morphine, daily NRS scores at rest and on movement for 3 days, the time to remove the nasogastric tube and urinary catheter, the time to commence fluid and soft diet, and lastly the length of hospital stay.

### 2.3. Statistics

The statistics were analyzed using SPSS Version 25 and R software. The continuous data were presented by mean ± standard deviation (Mean ± SD), and categorical variables as percentages. For statistical analysis, analysis of variance (ANOVA) was performed, and intra-group difference was examined by Tukey HSD post hoc analysis. Data were further adjusted for the presence of major resection, HCV infection, serum albumin, and hemoglobin levels using the PROC GLM procedure. Proportions were compared using the Chi-square test. For all statistical analysis, a *p*-value less than 0.05 was considered as statistically significant.

## 3. Results

In total, 564 patients received upper gastrointestinal surgeries and requested PCA during the study period; 293 patients underwent liver resections, of which 23 patients were excluded for having laparoscopic surgeries in this retrospective study, leaving a total of 243 patients for analysis (Figure 1). The study cohort had a mean age of 60.90 ± 11.07 years (68.7% male, 31.3% female). As shown in Table 1, the patients showed no statistically significant difference in their preexisting comorbidities such as the presence of diabetes mellitus, hypertension, end-stage renal disease (ESRD), cardiovascular disease (CAD), cerebrovascular accidents (CVA), cancer history or hepatitis B virus (HBV) infection; however, IV-PCA group had a higher proportion of hepatitis C virus infection (HCV). The three groups also showed no difference in histories of previous surgical procedures, cigarette smoking, or alcohol abuse. Except for higher baseline hemoglobin and albumin levels in the TEA group, which were subsequently adjusted for outcome analysis, the three groups showed no other baseline difference in indocyanine green retention test at 15 min ICG-15(%), platelet, alanine transaminases, alkaline phosphatase, total bilirubin, or alpha-fetoprotein levels. In Table 2, 107 of the 243 patients (44%) received major resections, defined as tri-segmentectomy, right/left lobectomy and extended right/left lobectomy, among which, a majority (51%) had IV-PCA + NBs (*p* = 0.0495). Indications for liver resection included primary hepatic cancer (n = 164, 67.5%), biliary origin carcinoma (*n* = 27, 11.1%), benign liver disease (*n* = 26, 10.7%), metastatic tumor from colon, ovarian, lung and renal (*n* = 23, 9.5%), sarcoma (*n* = 3, 1.2%). None of the intraoperative parameters, including operative time, intraoperative diagnosis, blood loss requiring blood transfusion, fluid supplementation, and urine output showed statistical significance. No pneumonia or unexpected ICU admission was documented during our data collection.

### 3.1. Primary Outcome

As shown in Table 3, after adjustment for perioperative factors, the total morphine consumption within the first three postoperative days was 83.53 ± 49.51, 64.17 ± 31.96, and 9.21 ± 4.91 mg in the IV-PCA, IV-PCA + NBs group and TEA group, respectively. Both the IV-PCA and IV-PCA + NBs groups required much more morphine postoperatively than TEA group, as expected. IV-PCA + NBs also required significantly less opioid than the IV-PCA group (*p* = 0.0008). The difference was further observed when body weight was taken into consideration (*p* < 0.0001) among the three groups. That said, no significance was observed for analgesic effects among all three modalities at rest and upon movement. Interestingly on day 3 on movement, a marginal significance (*p* = 0.0502) was observed when comparing IV-PCA + NBs to TEA. NRS scores for 3 consecutive postoperative days both at rest and upon movement are illustrated in Figure 2.

### 3.2. Secondary Outcome

Postoperative outcomes were summarized in Table 4. On average, the IV-PCA, IV-PCA + NBs, and TEA groups had their NG tubes removed on the postoperative day 1.35 ± 1.09, 1.29 ± 0.86, and 1.29 ± 0.7, respectively, with no statistical significance observed among the three groups. Patients, on average, had their urinary catheter removed on postoperative day 4.93 ± 5.08, 3.87 ± 1.31, and 3.70 ± 1.30, respectively. The time to remove the urinary catheter appeared significant comparing the three groups, in which the IV-PCA group had their catheters removed in the latest; the intra-group analysis showed statistical significance only between IV-PCA and IV-PCA + NBs (*p* = 0.0461) but not between IV-PCA + NBs and TEA groups. The return of bowel function was demonstrated as the return of bowel sounds during postoperative recovery, which then allowed a sip of water and a start of a soft diet. Patients, on average, started on fluid and soft diet intake on postoperative day 1 and 2, respectively, and no difference was observed among the three groups. The average length of hospital stay for all 3 groups was 11.79 ± 6.16 days. IV-PCA, IV-PCA + NBs, and TEA groups stayed for 12.71 ± 7.26, 11.79 ± 5.71, 10.02 ± 4.52 days, respectively (*p* < 0.0001). Intra-group significance was not observed between groups.

## 4. Discussion

In hepatobiliary patients, more than half suffered from substantial surgical wounds under laparotomy. The intense surgical pain over their right upper abdomen often discourages patients from deep breathing, coughing, eating, and moving normally, all of which could lead to undesirable complications such as pneumonia, urinary retention, and delayed return of bowel movement. Therefore, it is of utter importance that postoperative pain is well managed for the patients undergoing hepatectomy to speed the recovery from the surgery without compromising patient satisfaction [23,48]. Opioids have been the mainstay of postoperative pain control; nonetheless, opioid is notoriously associated with unwanted side effects with ceiling effect [47]. Epidural analgesia initially gained a fair reputation in pain management, especially in labor analgesia, and was gradually incorporated into the pain control of abdominal surgery. Patients with epidural analgesia usually require fewer opioids, experience less opioid-related side effects, and have a smaller subjective pain score [14,18,23]. Despite these advantages, it is alerted that patients who need liver resection are likely under multiple comorbidities, in particular, liver dysfunction, which could lead to coagulopathy and epidural hematoma [49]. Some may be vulnerable to severe hypotension due to the combined effects of sympathetic blockade followed by epidural analgesia and restricted fluid strategy during liver operation [23]. In addition, with the assistance of “loss of resistance” technique, up to 20% failure rate is still noted [50], and different anatomical characteristics of a patient could be challenging even to experienced anesthesiologists [51]. Due to these potential limitations and adverse events, the literature remains equipoised on whether PCEA was the preferred method for pain management and other analgesic techniques for pain management in open liver surgeries have been discussed [15,16,52,53,54]. For example, bilateral continuous paravertebral catheters administering local anesthetics have been used and shown as an effective adjunct for postoperative pain management [55,56]. A single dose of intrathecal morphine, in addition to other conventional analgesic strategies, has been shown to be safe and effective in providing postoperative analgesia when an epidural catheter is contraindicated [57,58].

With the concept of pain ladder management and multimodal analgesia protocol, IV-PCA constituted with opioids has been widely utilized in providing pain control; nevertheless, inadequate pain control is not unseen because of the side effects from opioids. In an attempt to provide adequate pain control without adverse effects, ultrasound-guided peripheral NBs appears promising in open liver resections [59,60]. RS and TAP nerve blocks using long-acting local anesthetics reduce total intravenous opioids requirement [61]. While local anesthetics-associated toxicity (LAST) is long debated as a major concern, few cases were reported in association with LAST in patients receiving abdominal wall NBs especially when ultrasound guidance is utilized. The current study, by comparing three different analgesic modalities, including IV-PCA plus abdominal wall NBs, attempts to suggest appropriate pain control following open liver resections.

In the current study, as the PCEA group did not receive a basal rate of opioid, it is expected that these patients would have the least opioid requirement. The comparable NRS with the least opioid consumption suggested TEA to be superior to IV-PCA for postoperative pain management, in consistency with the literature [15,16]. In terms of other postoperative outcomes, such as the time to remove urinary catheter and length of hospital stay, IV-PCA + NBs appeared clinically superior to IV-PCA alone, possibly attributed to the fact that less opioid was required in patients with nerve blocks. IV-PCA + NBs group appeared comparable to TEA when pain scores were kept below 5 on a scale of 10. No difference was observed in the time to remove nasogastric (NG) tubes or the time to start a fluid and soft diet between IV-PCA + NBs to TEA. It appears that IV-PCA + NBs delivered the benefits of adequate pain control without significantly delaying the return of bowel and urinary function, despite more opioid, is still required for IV-PCA + NBs than TEA. In consistency with the literature, IV-PCA + NBs warrants the highest efficacy for pain management and fewer side effects without the potential for the increased risk of epidural complications [62]. Therefore, a supplementation of ultrasound-guided abdominal wall NBs to IV-PCA postoperatively appears to act synergistically [20,21,25,63,64] and should be considered as a suitable alternative for pain management when TEA is contraindicated.

The advantages of our study are that instead of simple comparisons between two pain management options, we have assessed three postoperative pain control modalities for distressing pain after liver resections. To avoid biases from the surgical point of view, only patients of the same surgical team with similar surgical techniques and postoperative care practices were included in the study. In addition, the demographic characteristics and intraoperative management, including the requirement for blood transfusion and fluid administration, were comparable among the three groups. A group of anesthesiologists dedicated to the Acute Pain Service also provided postoperative pain management for all these patients. Patients were assessed separately by the nursing staff, yielding consistency in the personnel. Although the present study has established practical implications, it is not without limitations. The retrospective nature of the study precluded an equal number of patients in each group and the patients in the TEA group were much fewer than the others. Such discrepancy may be the result of the anesthesiologists’ preference. In addition, patients’ preference was taken into consideration, permitted by their clinical conditions when the type of postoperative analgesia was selected. Cost and ethnic disparities, therefore, limited the number of patients choosing TEA for postoperative pain control, further rendering an unequal sample size among the three groups. Future larger prospective studies are thus required to validate our results.

The aim of the study was not to dispute the use of TEA; in fact, we have demonstrated that IV-PCA, in addition to NBs, may be a reasonable alternative in providing adequate analgesia without significant opioid-related side effects in open liver resections. With the implementation of ultrasound guidance, anterior abdominal wall NBs can be a safe and practical technique in clinical use. But one must bear in mind that NBs only provide analgesia of the abdominal wall and not the abdominal viscera.

## 5. Conclusions

A combination of IV-PCA with abdominal wall NBs may effectively achieve a comparable analgesic effect to PCEA without causing a delay in the return of normal bowel and bladder function, or longer hospitalization postoperatively in patients undergoing liver resections. Given the relatively straightforward technique under ultrasound guidance, IV-PCA + NBs may be considered as a reasonable and reliable analgesic modality for patients undergoing open liver resections.

## Figures and Tables

**Figure 1 biomedicines-10-02411-f001:**
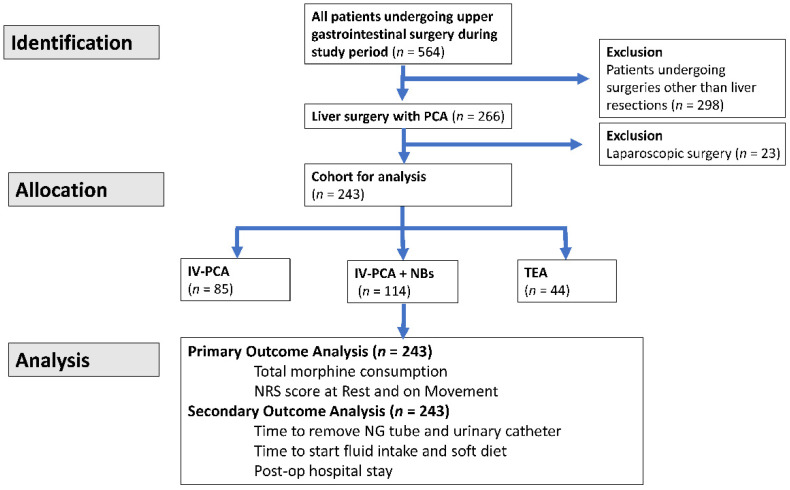
The flowchart of patient identification, allocation, and analysis.

**Figure 2 biomedicines-10-02411-f002:**
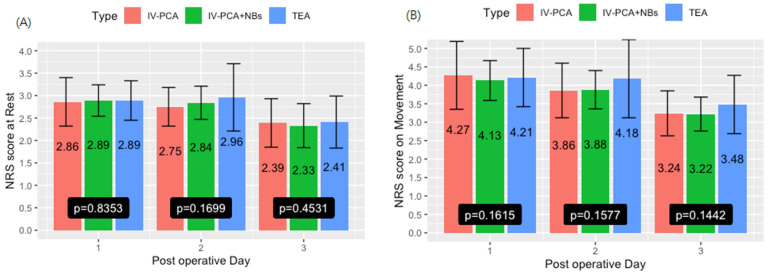
Numerical rating scale (NRS) scores on a scale of 10 at rest and on movement on the first three postoperative days (For better depiction, the vertical axis was drawn to up to 5 as no patients experienced NRS greater than 5). (**A**) NRS at rest on postoperative day 1, 2, and 3. (**B**) NRS on movement on postoperative day 1, 2, and 3.

**Table 1 biomedicines-10-02411-t001:** Clinical characteristics of patient (*n* = 243).

Variables	Total		IV-PCA (*n* = 85)		IV-PCA + NBs (*n* = 114)		TEA (*n* = 44)		*p*-Value
	No.	(%)	No.	(%)	No.	(%)	No.	(%)	
**Male gender**	167	68.7%	57	67.1%	78	68.4%	32	72.7%	0.802
**Comorbidity**									
** Diabetes**	66	27.2%	28	32.9%	25	21.9%	13	29.5%	0.208
** Hypertension**	105	43.2%	39	45.9%	51	44.7%	15	34.1%	0.397
** ESRD**	5	2.1%	1	1.2%	4	3.5%	0	0.0%	0.295
** CAD**	10	4.1%	3	3.5%	6	5.3%	1	2.3%	0.659
** CVA**	7	2.9%	4	4.7%	3	2.6%	0	0.0%	0.310
** Cancer history**	44	18.1%	11	12.9%	27	23.7%	6	13.6%	0.105
** HBV infection**	123	50.6%	39	45.9%	58	50.9%	26	59.1%	0.363
** HCV infection**	36	14.8%	17	20.0%	18	15.8%	1	2.3%	0.025
**Cigarette smoking**	101	41.6%	33	38.8%	52	45.6%	16	36.4%	0.467
**Alcohol consumption**	79	32.5%	29	34.1%	38	33.3%	12	27.3%	0.71
**Previous surgery**	41	16.9%	17	20.0%	15	13.2%	9	20.5%	0.347
	**Mean** **±** **SD**		**Mean** **±** **SD**		**Mean** **±** **SD**		**Mean** **±** **SD**		** *p* ** **-Value**
**Age (years)**	60.90 ± 11.07		62.34 ± 11.64		60.85 ± 10.75		58.25 ± 10.48		0.138
**Weight (kg)**	66.77 ± 11.99		66.22 ± 12.91		66.63 ± 10.89		68.20 ± 13.01		0.665
**ICG-15 (%)**	10.09 ± 7.27		10.66 ± 7.54		10.44 ± 7.62		7.86 ± 5.10		0.151
**Hemoglobin (g/dL)**	13.5 ± 1.81		13.67 ± 1.64		13.18 ± 1.96		14 ± 1.56		0.0209
**Albumin (g/dL)**	4.22 ± 0.41		4.14 ± 0.36		4.19 ± 0.44		4.44 ± 0.35		<0.001
**Platelet (1000/uL)**	194.07 ± 76.29		184.84 ± 73.96		202.82 ± 85.34		189.23 ± 50.33		0.233
**ALT (U/L)**	46.41 ± 49.47		48.73 ± 47.56		45.89 ± 55.04		43.27 ± 37.01		0.83
**Alkaline Phosphatase (U/L)**	90.42 ± 55.43		90.75 ± 42.82		95.16 ± 68.23		77.73 ± 35.62		0.21
**Total Bilirubin (mg/dL)**	0.74 ± 1.31		0.65 ± 0.24		0.85 ± 1.9		0.62 ± 0.25		0.454
**α-fetoprotein (ng/mL)**	3244.34 ± 19057.55		5958.48 ± 27930.21		2097.30 ± 13786.73		1034.08 ± 2917.73		0.28

ESRD, end-stage renal disease; CAD, coronary artery disease; CVA, cerebrovascular accident; HBV, hepatitis B virus; HCV, hepatitis C virus; ICG-15, indocyanine green retention test at 15 min; ALT, alanine aminotransferase; min, minutes; IV-PCA, intravenous patient controlled analgesia; NB, nerve block; SD, standard deviation; TEA, thoracic epidural analgesia.

**Table 2 biomedicines-10-02411-t002:** Perioperative parameters (*n* = 243).

Variables	Total		IV-PCA (*n* = 85)		IV-PCA + NBs (*n* = 114)		TEA (*n* = 44)		*p*-Value
	No.	(%)	No.	(%)	No.	(%)	No.	(%)	
**Intra-Abdominal Adhesion**	39	16.0%	16	18.8%	15	13.2%	8	18.2%	0.511
**Major Resections**	107	44%	36	42%	58	51%	13	30%	0.0495
**Intraoperative** **Blood Transfusion**	116	48%	46	54%	55	48%	15	34%	0.096
**Diagnosis**									0.316
HCC	164	67.49%	61	71.76%	76	66.67%	27	61.36%	
CCC	27	11.11%	8	9.41%	12	10.53%	7	15.91%	
Metastatic liver tumor	23	9.47%	9	10.59%	12	10.53%	2	4.55%	
Benign disease	26	10.70%	7	8.24%	11	9.65%	8	18.18%	
Sarcoma	3	1.23%	0	0.00%	3	2.63%	0	0.00%	
	**Mean** **±** **SD**		**Mean** **±** **SD**		**Mean** **±** **SD**		**Mean** **±** **SD**		** *p* ** **-Value**
**Operative Time (min)**	314.42 ± 105.69		315.51 ± 106.08		317.73 ± 102.23		303.77 ± 115.21		0.755
**Intraoperative** **Fluid Transfusion (mL)**	1537.62 ± 789.97		1641.06 ± 814.93		1457.46 ± 800.53		1545.45 ± 701.72		0.269
**Intraoperative** **Urine Output** **(mL/h/kilograms)**	1.16 ± 0.77		1.24 ± 0.79		1.09 ± 0.71		1.21 ± 0.89		0.3612
**Intraoperative** **Blood Loss (mL)**	512.88 ± 614.69		604.82 ± 719.57		450.44 ± 532.21		497.05 ± 585.95		0.212

HCC, hepatocellular carcinoma; CCC, cholangiocellular carcinoma; min, minutes; IV-PCA, intravenous patient-controlled analgesia; NB, nerve block; SD, deviation; TEA, thoracic epidural analgesia.

**Table 3 biomedicines-10-02411-t003:** Primary outcomes.

Variables	Total	IV-PCA (*n* = 85)	IV-PCA + NBs (*n* = 114)	TEA (*n* = 44)	*p*-Value			
	Mean ± SD					IV-PCA vs. IV-PCA + NBs	IV-PCA vs. TEA	IV-PCA + NBs vs. TEA
**Total Morphine consumption (mg)**	60.99 ± 44.75	83.53 ± 49.51	64.17 ± 31.96	9.21 ± 4.91	<0.001	0.0008	<0.001	<0.001
**Total Morphine use/weight (mg/kg)**	0.92 ± 0.64	1.26 ± 0.67	0.96 ± 0.46	0.14 ± 0.07	<0.0001	0.0002	<0.0001	<0.0001
**Day 1 rest NRS**	2.88 ± 0.44	2.86 ± 0.54	2.89 ± 0.35	2.89 ± 0.44	0.8353	0.8753	0.9020	0.9968
**Day 2 rest NRS**	2.83 ± 0.48	2.75 ± 0.43	2.84 ± 0.37	2.96 ± 0.75	0.1699	0.3298	0.0780	0.4637
**Day 3 rest NRS**	2.36 ± 0.52	2.39 ± 0.54	2.33 ± 0.49	2.41 ± 0.58	0.4531	0.6222	0.9537	0.5628
**Day 1 move NRS**	4.19 ± 0.74	4.27 ± 0.92	4.13 ± 0.54	4.21 ± 0.79	0.1615	0.3790	0.7138	0.9775
**Day 2 move NRS**	3.93 ± 0.73	3.86 ± 0.74	3.88 ± 0.52	4.18 ± 1.06	0.1577	0.9928	0.1179	0.1152
**Day 3 move NRS**	3.27 ± 0.59	3.24 ± 0.61	3.22 ± 0.46	3.48 ± 0.79	0.1442	0.9402	0.1145	0.0502

IV-PCA, intravenous patient-controlled analgesia; NBs, nerve blocks; TEA, thoracic epidural analgesia; NRS, numerical rating scale; SD, standard deviation.

**Table 4 biomedicines-10-02411-t004:** Secondary outcomes.

Variables	Total	IV-PCA (*n* = 85)	IV-PCA + NBs (*n* = 114)	TEA (*n* = 44)	*p*-Value			
	Mean ± SD					IV-PCA vs. IV-PCA + NBs	IV-PCA vs. TEA	IV-PCA + NBs vs. TEA
**NG tube removal (days)**	1.32 ± 0.92	1.3 5 ± 1.09	1.29 ± 0.86	1.29 ± 0.7	0.9430	0.9033	0.9825	0.9888
**Urinary catheter removal (days)**	4.21 ± 3.21	4.93 ± 5.08	3.87 ± 1.31	3.70 ± 1.30	0.0414	0.0461	0.2281	0.9931
**Start on sip of water (days)**	1.42 ± 0.96	1.51 ± 1.2	1.38 ± 0.86	1.36 ± 0.68	0.1933	0.5913	0.7635	0.9998
**Start on soft diet (days)**	2.81 ± 1.98	2.85 ± 1.96	2.85 ± 2.18	2.66 ± 1.42	0.1130	0.9998	0.9998	0.8497
**Postoperative hospital stay(days)**	11.79 ± 6.16	12.71 ± 7.26	11.79 ± 5.71	10.02 ± 4.52	<.0001	0.5602	0.5724	0.9613

IV-PCA, intravenous patient-controlled analgesia; NBs, nerve blocks; TEA, thoracic epidural analgesia; NG, nasogastric; IV-PCA, intravenous patient-controlled analgesia; NB, nerve block; SD: standard deviation; TEA, thoracic epidural analgesia.

## Data Availability

All data generated or analyzed during the study are included in this published article. Raw data may be requested from the authors with the permission of the institution.

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
