# Peer review of "A Retrospective Comparison of Three Patient-Controlled Analgesic Strategies: Intravenous Opioid Analgesia Plus Abdominal Wall Nerve Blocks versus Epidural Analgesia versus Intravenous Opioid Analgesia Alone in Open Liver Surgery"

_biomedicines, 2022, doi:10.3390/biomedicines10102411_

Round 1
Reviewer 1 Report (Previous Reviewer 2)
Authors were attentive to revewer's suggestions and addressed them thoughtfully. My only remaining recommendation is to provide the concentrations of medications used IV-PCA as only the volumes were provided (i.e., mcg/ml). The concentrations and/or total opioid dose is more important to include that the volume.
Author Response
Please see the attachment.

Reviewer 2 Report (New Reviewer)
I would like to thank the authors for their work.
I think that the study is adequately presented such as discussion and conclusion.
Also the limits of this retrospective analysis are well presented.
I have just two questions:
I do not understand if patients with TEA used opioids.
How did the authors set PCA ? (they indicate morphine and fentanyl, but using which concentrations?)
Author Response
Please see the attachment.

This manuscript is a resubmission of an earlier submission. The following is a list of the peer review reports and author responses from that submission.
Round 1
Reviewer 1 Report
Tsai and colleagues have performed a retrospective case series of patients having a hepatectomy and focused on outcomes related to analgesic method. The study is relatively well presented in terms of structure, diagrams and tables. Despite presenting the findings there appears to be a lack of clarity on what this study contributes to the literature. If a major revision is possible the authors will need to focus more on what their study adds/contributes. The introduction and discussion need considerable attention and more scientific dialogue is required. Within the study there are many variables mentioned as important in the intro and discussion and not included within the study including pneumonia, return to bowel function, hypotension, post-operative morbidity. These will all need to be included if this article is to be published. It contributes very little in its current format.
Major change: Need to add data on pneumonia, return to bowel function, hypotension, post-operative morbidity, time in ICU/HDU, readmission to ICU/HDU
I have added additional points per section below:
Abstract:
· You need to mention what kind of study this is in the abstract, is it retrospective, prospective, randomised, adequately powered?
· You should not use words like substantially and mostly similar. You need to write exact figures even in the abstract
· Why have you chosen morphine requirement as the first part of your results, what does this mean?
· You need to include numbers here, please bear in mind 99% of people will only read abstract- you need to give key message with figures
Introduction:
· Remove distressing cutting wound- it sounds unscientific- just label the incision type
· Change ‘most often took’ to something like “ the Makuuchi incision is the most frequently used surgical incision
· Instead of saying no gold standard it is better to state there is clinical equipoise
· TEA rarely have severe complications, you need to explain this better. Failure rate and maintenance are the main issues , the bring in hypotension and need for vasopressors etc and then lastly state severe complications
· When is TEA contraindicated?
· Overall you need to give your audience more info on figures from other studies, a lot of the statements are too vague, you need to outline things with quotes of data from other studies. Writing statement is not good enough.
Methods:
· You will need to describe better why you chose the patients for the analgesic regimen. This is extremely important. Do you have a protocol, is it dependent on anaesthesiologist?
· Flowchast needs to be clearer that this was retrospective as it may be mistaken as prospective, you do not enrol for a retrospective study.
· Why have you not chosen to include patients with laparoscopic resection and do a mini comparison?
· Why were you using a basal rate on PCA? This is not common, please reference rationale why, is this based on data from your institution?
· Why were catheters not used in patient with nerve blocks?
· Why are you using morphine in these patients? Is it not dangerous based on what you mentioned in the introduction based on metabolism?
· Your outcomes are rudimentary, what about pneumonia and other complications, ICU or HDU length of stay
· You need to bear in mind opioid requirements mean nothing unless you look at opioid related side effects
Results:
· Why did the PCA group have a higher incidence of Hep C? Will this bias your results ?
· “Although statistically significant, all patients had hemoglobin and albumin levels within normal range” This sentence does not make sense
· There is also significance for major resections and the rationale and how this effects results will need to be discussed
· Why is the primary outcome opioid requirements? What does this mean? Although used in many old studies it means nothing unless you report on opioid related side effects. I would suggest length of stay or time in ICU/HDU a much more important clinical outcome. Pain scores would also be more acceptable as a key outcome. Opioid requirements should be near the bottom unless you can justify.
· Fig 2 should have a scale to 10
· You need to include data on pneumonia, readmission to ICU/HDU and other morbidities here
· You have also mentioned many issues with each analgesic regimen in the introduction but not included many in your results
Discussion:
· Overall the discussion needs to be improved, you need to again compare your findings of your study to the literature and what does it add?
· You should start out with key findings, what your study has added to the available literature and how it compares to other studies. Why have you started to talk about laparoscopic surgery when you excluded this?
· You need to make comments on the significant differences on the patient cohorts in your results and how this has effected your results
· Again, you have highlighted return to bowel movement and pneumonia and made no mention in you study, this looks poor
· Where is your data on hypotension since you mentioned again?
· What are the next steps ? A RCT
· How has this impacted on your practice? Do all patients get NB now?
· What about the use of catheters?
Conclusion:
· Where have you demonstrated a return to bowel function
· You need to make a more definitive statement on your conclusion , you have somehow suggested everyone should have PCEA unless not applicable??
Reviewer 2 Report
This is a retrospective study that evaluates pain-related outcomes in patients undergoing open liver resection for 3 common treatment outcomes. Overall, the study is relatively simple and does provide data necessary to evaluate best clinical practice so it would be of value to those providing pain control in this setting.
The introduction provides a detailed description of analgesic options for these patients with a nice explanations of the advantages and disadvantages of each as it relates specifically to liver resections. The methods are adequately detailed with minor exceptions described below. The results include thoughtfully chosen outcomes. However, the presentation of the values, specifically the variability should be improved (see below).
The discussion of this manuscript is the clear weakness and should be greatly improved. One paragraph toward the end of the section simply restates the overall findings of the study and there is no incorporation of how their findings compare to prior research. For example, the authors found increased time to urinary catheter removal with the IV PCA group but not when it was combined with the nerve block. Reasons for measuring time to catheter removal and whether prior studies have shown an impact on time to removal and post-operative pain control would be more appropriate to include. Why nerve block may have influenced this outcome would also be appropriate and is missing. Instead, the discussion often discusses issues that are not being evaluated in the study. For example, reasons that liver resection cannot be performed laparoscopically does not provide any context for the comparison of TEA to IV PCA for pain control and is not relevant.
Other minor comments:
1. As this is a retrospective study where groups were not randomly assigned, a discussion of the biases involved in selection for one of the 3 analgesic options is needed as factors that were associated with a patient receiving TEA may also be associated with the pain-related outcomes assessed. This was not addressed.
2. The dosages of IV-PCA were not provided – just the volumes. The concentrations and/or total opioid dose is more important to include.
3. Since TEA was the only analgesic option where the patient didn’t receive a basal rate of opioid, it is expected that these patients would have lower opioid consumption. This should be mentioned. the amount of opioid consumption above the basal dosage may be interesting to see between groups, if this is available.
4. In addition to the means, standard deviation or other measure of variance should also be included. SEM is included on Table 3 as an abbreviation, but it is not clear where these values are. In addition, SD should be used as the population is not being measured here.
5. The table 3 should include an additional significant digit
6. Figure 2 has no error bars. Since the p value doesn’t =1, the sd is > 0. Again, since NRS is a mean, the values should be taken out to at least an additional significant digit to show the variability between groups.
7. The paper should be edited by a native English speaker. There is use of words in unexpected, but not necessarily inappropriate ways that should be corrected. This didn’t detract from my understanding, but it was distracting at times and interfered with the readability.
Round 2
Reviewer 1 Report
The authors need to answer the questions individually, there is no evidence that they have answered any appropriately. Bullet points are not acceptable and this should have been checked before sending back to me.